# Subset Selection under Noise

**Chao Qian**[1]      **Jing-Cheng Shi**[2]      **Yang Yu**[2]      **Ke Tang**[3,1]      **Zhi-Hua Zhou**[2]

[1]Anhui Province Key Lab of Big Data Analysis and Application, USTC, China
[2]National Key Lab for Novel Software Technology, Nanjing University, China
[3]Shenzhen Key Lab of Computational Intelligence, SUSTech, China

chaoqian@ustc.edu.cn   tangk3@sustc.edu.cn
{shijc,yuy,zhouzh}@lamda.nju.edu.cn

## Abstract

The problem of selecting the best $k$-element subset from a universe is involved in many applications. While previous studies assumed a noise-free environment or a noisy monotone submodular objective function, this paper considers a more realistic and general situation where the evaluation of a subset is a noisy monotone function (not necessarily submodular), with both multiplicative and additive noises. To understand the impact of the noise, we firstly show the approximation ratio of the greedy algorithm and POSS, two powerful algorithms for noise-free subset selection, in the noisy environments. We then propose to incorporate a noise-aware strategy into POSS, resulting in the new PONSS algorithm. We prove that PONSS can achieve a better approximation ratio under some assumption such as i.i.d. noise distribution. The empirical results on influence maximization and sparse regression problems show the superior performance of PONSS.

## 1   Introduction

Subset selection is to select a subset of size at most $k$ from a total set of $n$ items for optimizing some objective function $f$, which arises in many applications, such as maximum coverage [10], influence maximization [16], sparse regression [17], ensemble pruning [23], etc. Since it is generally NP-hard [7], much effort has been devoted to the design of polynomial-time approximation algorithms.

The greedy algorithm is most favored for its simplicity, which iteratively chooses one item with the largest immediate benefit. Despite the greedy nature, it can perform well in many cases. For a monotone submodular objective function $f$, it achieves the $(1 - 1/e)$-approximation ratio, which is optimal in general [18]; for sparse regression where $f$ can be non-submodular, it has the best-so-far approximation bound $1 - e^{-\gamma}$ [6], where $\gamma$ is the submodularity ratio.

Recently, a new approach Pareto Optimization for Subset Selection (POSS) has been shown superior to the greedy algorithm [21, 24]. It reformulates subset selection with two simultaneous objectives, i.e., optimizing the given objective and minimizing the subset size, and employs a randomized iterative algorithm to solve this bi-objective problem. POSS is proved to achieve the same general approximation guarantee as the greedy algorithm, and is shown better on some subclasses [5]. The Pareto optimization method has also been successfully applied to solve subset selection with general cost constraints [20] as well as ratio optimization of monotone set functions [22].

Most of the previous studies assumed that the objective function is noise-free. However, we can only have a noisy evaluation in many realistic applications. For examples, for influence maximization, computing the influence spread objective is #P-hard [2], and thus is often estimated by simulating the random diffusion process [16], which brings noise; for sparse regression, only a set of limited data can be used for evaluation, which makes the evaluation noisy; and more examples include maximizing information gain in graphical models [4], crowdsourced image collection summarization [26], etc.

To the best of our knowledge, only a few studies addressing noisy subset selection have been reported, which assumed monotone submodular objective functions. Under the general multiplicative noise model (i.e., the noisy objective value $F(X)$ is in the range of $(1 \pm \epsilon)f(X)$), it was proved that no polynomial-time algorithm can achieve a constant approximation ratio for any $\epsilon > 1/\sqrt{n}$, while the greedy algorithm can achieve a $(1 - 1/e - 16\delta)$-approximation ratio for $\epsilon = \delta/k$ as long as $\delta < 1$ [14]. By assuming that $F(X)$ is a random variable (i.e., random noise) and the expectation of $F(X)$ is the true value $f(X)$, it was shown that the greedy algorithm can achieve nearly a $(1 - 1/e)$-approximation guarantee via uniform sampling [16] or adaptive sampling [26]. Recently, Hassidim and Singer [13] considered the consistent random noise model, where for each subset $X$, only the first evaluation is a random draw from the distribution of $F(X)$ and the other evaluations return the same value. For some classes of noise distribution, they provided polynomial-time algorithms with constant approximations.

In this paper, we consider a more general situation, i.e., noisy subset selection with a monotone objective $f$ (not necessarily submodular), for both multiplicative noise and additive noise (i.e., $F(X)$ is in the range of $f(X) \pm \epsilon$) models. The main results are:

• Firstly, we extend the approximation ratio of the greedy algorithm from the submodular case [14] to the general situation (**Theorems 1, 2**), and also slightly improve it.

• Secondly, we prove that the approximation ratio of POSS is nearly the same as that of the greedy algorithm (**Theorems 3, 4**). Moreover, on two maximum coverage cases, we show that POSS can have a better ability of avoiding the misleading search direction due to the noise (**Propositions 1, 2**).

• Thirdly, we introduce a noise-aware comparison strategy into POSS, and propose the new PONSS algorithm for noisy subset selection. When comparing two solutions with close noisy objective values, POSS selects the solution with the better observed value, while PONSS keeps both of them such that the risk of deleting a good solution is reduced. With some assumption such as i.i.d. noise distribution, we prove that PONSS can obtain a $\frac{1-\epsilon}{1+\epsilon}(1 - e^{-\gamma})$-approximation ratio under multiplicative noise (**Theorem 5**). Particularly for the submodular case (i.e., $\gamma = 1$) and $\epsilon$ being a constant, PONSS has a constant approximation ratio. Note that for the greedy algorithm and POSS under general multiplicative noise, they only guarantee a $\Theta(1/k)$ approximation ratio. We also prove the approximation ratio of PONSS under additive noise (**Theorem 6**).

We have conducted experiments on influence maximization and sparse regression problems, two typical subset selection applications with the objective function being submodular and non-submodular, respectively. The results on real-world data sets show that POSS is better than the greedy algorithm in most cases, and PONSS clearly outperforms POSS and the greedy algorithm.

We start the rest of the paper by introducing the noisy subset selection problem. We then present in three subsequent sections the theoretical analyses for the greedy, POSS and PONSS algorithms, respectively. We further empirically compare these algorithms. The final section concludes this paper.

## 2 Noisy Subset Selection

Given a finite nonempty set $V = \{v_1, \ldots, v_n\}$, we study the functions $f : 2^V \to \mathbb{R}$ defined on subsets of $V$. The subset selection problem as presented in Definition 1 is to select a subset $X$ of $V$ such that a given objective $f$ is maximized with the constraint $|X| \leq k$, where $|\cdot|$ denotes the size of a set. Note that we only consider maximization since minimizing $f$ is equivalent to maximizing $-f$.

**Definition 1** (Subset Selection). *Given all items $V = \{v_1, \ldots, v_n\}$, an objective function $f$ and a budget $k$, it is to find a subset of at most $k$ items maximizing $f$, i.e.,*

$$\arg\max_{X \subseteq V} f(X) \quad s.t. \quad |X| \leq k. \tag{1}$$

A set function $f : 2^V \to \mathbb{R}$ is monotone if for any $X \subseteq Y$, $f(X) \leq f(Y)$. In this paper, we consider monotone functions and assume that they are normalized, i.e., $f(\emptyset) = 0$. A set function $f : 2^V \to \mathbb{R}$ is submodular if for any $X \subseteq Y$, $f(Y) - f(X) \leq \sum_{v \in Y \setminus X}(f(X \cup \{v\}) - f(X))$ [19]. The submodularity ratio in Definition 2 characterizes how close a set function $f$ is to submodularity. It is easy to see that $f$ is submodular iff $\gamma_{X,k}(f) = 1$ for any $X$ and $k$. For some concrete non-submodular applications, bounds on $\gamma_{X,k}(f)$ were derived [1, 9]. When $f$ is clear, we will use $\gamma_{X,k}$ shortly.

---
**Algorithm 1** Greedy Algorithm
---
**Input**: all items $V = \{v_1, \dots, v_n\}$, a noisy objective function $F$, and a budget $k$
**Output**: a subset of $V$ with $k$ items
**Process**:

  1: Let $i = 0$ and $X_i = \emptyset$.
  2: **repeat**
  3:    Let $v^* = \arg\max_{v \in V \setminus X_i} F(X_i \cup \{v\})$.
  4:    Let $X_{i+1} = X_i \cup \{v^*\}$, and $i = i + 1$.
  5: **until** $i = k$
  6: **return** $X_k$
---

**Definition 2** (Submodularity Ratio [6])**.** *Let $f$ be a non-negative set function. The submodularity ratio of $f$ with respect to a set $X$ and a parameter $k \geq 1$ is*

$$\gamma_{X,k}(f) = \min_{L \subseteq X, S:|S| \leq k, S \cap L = \emptyset} \frac{\sum_{v \in S} \big(f(L \cup \{v\}) - f(L)\big)}{f(L \cup S) - f(L)}.$$

In many applications of subset selection, we cannot obtain the exact objective value $f(X)$, but rather only a noisy one $F(X)$. In this paper, we will study the multiplicative noise model, i.e.,

$$(1 - \epsilon)f(X) \leq F(X) \leq (1 + \epsilon)f(X), \tag{2}$$

as well as the additive noise model, i.e.,

$$f(X) - \epsilon \leq F(X) \leq f(X) + \epsilon. \tag{3}$$

## 3   The Greedy Algorithm

The greedy algorithm as shown in Algorithm 1 iteratively adds one item with the largest $F$ improvement until $k$ items are selected. It can achieve the best approximation ratio for many subset selection problems without noise [6, 18]. However, its performance for noisy subset selection was not theoretically analyzed until recently. Let $OPT = \max_{X:|X| \leq k} f(X)$ denote the optimal function value of Eq. (1). Horel and Singer [14] proved that for subset selection with submodular objective functions under the multiplicative noise model, the greedy algorithm finds a subset $X$ with

$$f(X) \geq \frac{\frac{1-\epsilon}{1+\epsilon}}{1 + \frac{4k\epsilon}{(1-\epsilon)^2}} \left(1 - \left(\frac{1-\epsilon}{1+\epsilon}\right)^{2k} \left(1 - \frac{1}{k}\right)^k\right) \cdot OPT. \tag{4}$$

Note that their original bound in Theorem 5 of [14] is w.r.t. $F(X)$ and we have switched to $f(X)$ by multiplying a factor of $\frac{1-\epsilon}{1+\epsilon}$ according to Eq. (2).

By extending their analysis with the submodularity ratio, we prove in Theorem 1 the approximation bound of the greedy algorithm for the objective $f$ being not necessarily submodular. Note that their analysis is based on an inductive inequality on $F$, while we directly use that on $f$, which brings a slight improvement. For the submodular case, $\gamma_{X,k} = 1$ and the bound in Theorem 1 changes to be

$$f(X) \geq \frac{\frac{1-\epsilon}{1+\epsilon}\frac{1}{k}}{1 - \frac{1-\epsilon}{1+\epsilon}\left(1 - \frac{1}{k}\right)} \left(1 - \left(\frac{1-\epsilon}{1+\epsilon}\right)^k \left(1 - \frac{1}{k}\right)^k\right) \cdot OPT.$$

Comparing with that (i.e., Eq. (4)) in [14], our bound is tighter, since

$$\frac{1 - \left(\frac{1-\epsilon}{1+\epsilon}\right)^k \left(1 - \frac{1}{k}\right)^k}{1 - \frac{1-\epsilon}{1+\epsilon}\left(1 - \frac{1}{k}\right)} = \sum_{i=0}^{k-1} \left(\frac{1-\epsilon}{1+\epsilon}\left(1 - \frac{1}{k}\right)\right)^i \geq \sum_{i=0}^{k-1} \left(\left(\frac{1-\epsilon}{1+\epsilon}\right)^2 \left(1 - \frac{1}{k}\right)\right)^i \geq \frac{1 - \left(\frac{1-\epsilon}{1+\epsilon}\right)^{2k} \left(1 - \frac{1}{k}\right)^k}{1 + \frac{4k\epsilon}{(1-\epsilon)^2}} \cdot k.$$

Due to space limitation, the proof of Theorem 1 is provided in the supplementary material. We also show in Theorem 2 the approximation ratio under additive noise. The proof is similar to that of Theorem 1, except that Eq. (3) is used instead of Eq. (2) for comparing $f(X)$ with $F(X)$.

**Algorithm 2** POSS Algorithm
---
**Input**: all items $V = \{v_1, \ldots, v_n\}$, a noisy objective function $F$, and a budget $k$
**Parameter**: the number $T$ of iterations
**Output**: a subset of $V$ with at most $k$ items
**Process**:
 1: Let $\boldsymbol{x} = \{0\}^n$, $P = \{\boldsymbol{x}\}$, and let $t = 0$.
 2: **while** $t < T$ **do**
 3:    Select $\boldsymbol{x}$ from $P$ uniformly at random.
 4:    Generate $\boldsymbol{x}'$ by flipping each bit of $\boldsymbol{x}$ with probability $\frac{1}{n}$.
 5:    **if** $\nexists \boldsymbol{z} \in P$ such that $\boldsymbol{z} \succ \boldsymbol{x}'$ **then**
 6:       $P = (P \setminus \{\boldsymbol{z} \in P \mid \boldsymbol{x}' \succeq \boldsymbol{z}\}) \cup \{\boldsymbol{x}'\}$.
 7:    **end if**
 8:    $t = t + 1$.
 9: **end while**
10: **return** $\arg\max_{\boldsymbol{x} \in P, |\boldsymbol{x}| \leq k} F(\boldsymbol{x})$
---

**Theorem 1.** *For subset selection under multiplicative noise, the greedy algorithm finds a subset $X$ with*

$$f(X) \geq \frac{\frac{1-\epsilon}{1+\epsilon} \frac{\gamma_{X,k}}{k}}{1 - \frac{1-\epsilon}{1+\epsilon}\left(1 - \frac{\gamma_{X,k}}{k}\right)}\left(1 - \left(\frac{1-\epsilon}{1+\epsilon}\right)^k \left(1 - \frac{\gamma_{X,k}}{k}\right)^k\right) \cdot OPT.$$

**Theorem 2.** *For subset selection under additive noise, the greedy algorithm finds a subset $X$ with*

$$f(X) \geq \left(1 - \left(1 - \frac{\gamma_{X,k}}{k}\right)^k\right) \cdot \left(OPT - \frac{2k\epsilon}{\gamma_{X,k}}\right).$$

## 4 The POSS Algorithm

Let a Boolean vector $\boldsymbol{x} \in \{0,1\}^n$ represent a subset $X$ of $V$, where $x_i = 1$ if $v_i \in X$ and $x_i = 0$ otherwise. The Pareto Optimization method for Subset Selection (POSS) [24] reformulates the original problem Eq. (1) as a bi-objective maximization problem:

$$\arg\max_{\boldsymbol{x} \in \{0,1\}^n} \ (f_1(\boldsymbol{x}), \ f_2(\boldsymbol{x})), \quad \text{where } f_1(\boldsymbol{x}) = \begin{cases} -\infty, & |\boldsymbol{x}| \geq 2k \\ F(\boldsymbol{x}), & \text{otherwise} \end{cases}, \quad f_2(\boldsymbol{x}) = -|\boldsymbol{x}|.$$

That is, POSS maximizes the original objective and minimizes the subset size simultaneously. Note that setting $f_1$ to $-\infty$ is to exclude overly infeasible solutions. We will not distinguish $\boldsymbol{x} \in \{0,1\}^n$ and its corresponding subset for convenience.

In the bi-objective setting, the domination relationship as presented in Definition 3 is used to compare two solutions. For $|\boldsymbol{x}| < 2k$ and $|\boldsymbol{y}| \geq 2k$, it trivially holds that $\boldsymbol{x} \succeq \boldsymbol{y}$. For $|\boldsymbol{x}|, |\boldsymbol{y}| < 2k$, $\boldsymbol{x} \succeq \boldsymbol{y}$ if $F(\boldsymbol{x}) \geq F(\boldsymbol{y}) \wedge |\boldsymbol{x}| \leq |\boldsymbol{y}|$; $\boldsymbol{x} \succ \boldsymbol{y}$ if $\boldsymbol{x} \succeq \boldsymbol{y}$ and $F(\boldsymbol{x}) > F(\boldsymbol{y}) \vee |\boldsymbol{x}| < |\boldsymbol{y}|$.

**Definition 3** (Domination). *For two solutions $\boldsymbol{x}$ and $\boldsymbol{y}$,*
- *$\boldsymbol{x}$ weakly dominates $\boldsymbol{y}$ (denoted as $\boldsymbol{x} \succeq \boldsymbol{y}$) if $f_1(\boldsymbol{x}) \geq f_1(\boldsymbol{y}) \wedge f_2(\boldsymbol{x}) \geq f_2(\boldsymbol{y})$;*
- *$\boldsymbol{x}$ dominates $\boldsymbol{y}$ (denoted as $\boldsymbol{x} \succ \boldsymbol{y}$) if $\boldsymbol{x} \succeq \boldsymbol{y}$ and $f_1(\boldsymbol{x}) > f_1(\boldsymbol{y}) \vee f_2(\boldsymbol{x}) > f_2(\boldsymbol{y})$.*

POSS as described in Algorithm 2 uses a randomized iterative procedure to optimize the bi-objective problem. It starts from the empty set $\{0\}^n$ (line 1). In each iteration, a new solution $\boldsymbol{x}'$ is generated by randomly flipping bits of an archived solution $\boldsymbol{x}$ selected from the current $P$ (lines 3-4); if $\boldsymbol{x}'$ is not dominated by any previously archived solution (line 5), it will be added into $P$, and meanwhile those solutions weakly dominated by $\boldsymbol{x}'$ will be removed (line 6). After $T$ iterations, the solution with the largest $F$ value satisfying the size constraint in $P$ is selected (line 10).

In [21, 24], POSS using $\mathbb{E}[T] \leq 2ek^2n$ was proved to achieve the same approximation ratio as the greedy algorithm for subset selection without noise, where $\mathbb{E}[T]$ denotes the expected number of iterations. However, its approximation performance under noise is not known. Let $\gamma_{\min} = \min_{X:|X|=k-1} \gamma_{X,k}$. We first show in Theorem 3 the approximation ratio of POSS under multiplicative noise. The proof is provided in the supplementary material due to space limitation. The approximation ratio of POSS under additive noise is shown in Theorem 4, the proof of which is similar to that of Theorem 3 except that Eq. (3) is used instead of Eq. (2).

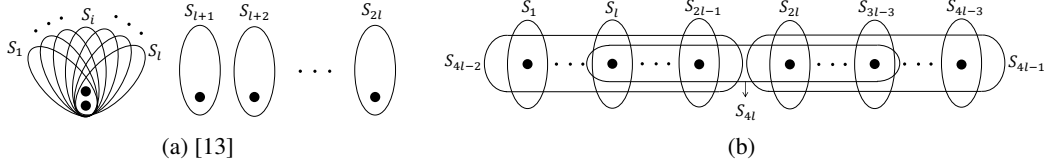

(a) [13]                                    (b)

Figure 1: Two examples of the maximum coverage problem.

**Theorem 3.** *For subset selection under multiplicative noise, POSS using $\mathbb{E}[T] \leq 2ek^2n$ finds a subset $X$ with $|X| \leq k$ and*

$$f(X) \geq \frac{\frac{1-\epsilon}{1+\epsilon} \frac{\gamma_{\min}}{k}}{1 - \frac{1-\epsilon}{1+\epsilon}\left(1 - \frac{\gamma_{\min}}{k}\right)}\left(1 - \left(\frac{1-\epsilon}{1+\epsilon}\right)^k \left(1 - \frac{\gamma_{\min}}{k}\right)^k\right) \cdot OPT.$$

**Theorem 4.** *For subset selection under additive noise, POSS using $\mathbb{E}[T] \leq 2ek^2n$ finds a subset $X$ with $|X| \leq k$ and*

$$f(X) \geq \left(1 - \left(1 - \frac{\gamma_{\min}}{k}\right)^k\right) \cdot \left(OPT - \frac{2k\epsilon}{\gamma_{\min}}\right) - \left(1 - \frac{\gamma_{\min}}{k}\right)^k \epsilon.$$

By comparing Theorem 1 with 3, we find that the approximation bounds of POSS and the greedy algorithm under multiplicative noise are nearly the same. Particularly, for the submodular case (where $\gamma_{X,k} = 1$ for any $X$ and $k$), they are exactly the same. Under additive noise, their approximation bounds (i.e., Theorems 2 and 4) are also nearly the same, since the additional term $(1 - \frac{\gamma_{\min}}{k})^k \epsilon$ in Theorem 4 can almost be omitted compared with other terms.

To further investigate the performances of the greedy algorithm and POSS, we compare them on two maximum coverage examples with noise. Maximum coverage as in Definition 4 is a classic subset selection problem. Given a family of sets that cover a universe of elements, the goal is to select at most $k$ sets whose union is maximal. For Examples 1 and 2, the greedy algorithm easily finds an optimal solution if without noise, but can only guarantee nearly a $2/k$ and $3/4$-approximation under noise, respectively. We prove in Propositions 1 and 2 that POSS can avoid the misleading search direction due to noise through multi-bit search and backward search, respectively, and find an optimal solution. Note that the greedy algorithm can only perform single-bit forward search. Due to space limitation, the proofs are provided in the supplementary material.

**Definition 4** (Maximum Coverage). *Given a ground set $U$, a collection $V = \{S_1, S_2, \ldots, S_n\}$ of subsets of $U$, and a budget $k$, it is to find a subset of $V$ (represented by $\boldsymbol{x} \in \{0,1\}^n$) such that*

$$\arg\max_{\boldsymbol{x} \in \{0,1\}^n} f(\boldsymbol{x}) = |\bigcup_{i:x_i=1} S_i| \quad s.t. \quad |\boldsymbol{x}| \leq k.$$

**Example 1.** *[13] As shown in Figure 1(a), $V$ contains $n = 2l$ subsets $\{S_1, \ldots, S_{2l}\}$, where $\forall i \leq l$, $S_i$ covers the same two elements, and $\forall i > l$, $S_i$ covers one unique element. The objective evaluation is exact except that $\forall \emptyset \subset X \subseteq \{S_1, \ldots, S_l\}, i > l$, $F(X) = 2 + \delta$ and $F(X \cup \{S_i\}) = 2$, where $0 < \delta < 1$. The budget satisfies that $2 < k \leq l$.*

**Proposition 1.** *For Example 1, POSS using $\mathbb{E}[T] = O(kn \log n)$ finds an optimal solution, while the greedy algorithm cannot.*

**Example 2.** *As shown in Figure 1(b), $V$ contains $n = 4l$ subsets $\{S_1, \ldots, S_{4l}\}$, where $\forall i \leq 4l - 3$: $|S_i| = 1$, $|S_{4l-2}| = 2l - 1$, and $|S_{4l-1}| = |S_{4l}| = 2l - 2$. The objective evaluation is exact except that $F(\{S_{4l}\}) = 2l$. The budget $k = 2$.*

**Proposition 2.** *For Example 2, POSS using $\mathbb{E}[T] = O(n)$ finds the optimal solution $\{S_{4l-2}, S_{4l-1}\}$, while the greedy algorithm cannot.*

## 5  The PONSS Algorithm

POSS compares two solutions based on the domination relation as shown in Definition 3. This may be not robust to noise, because a worse solution can appear to have a better $F$ value and then survive to replace the true better solution. Inspired by the noise handling strategy threshold selection [25], we modify POSS by replacing domination with $\theta$-domination, where $\boldsymbol{x}$ is better than $\boldsymbol{y}$ if $F(\boldsymbol{x})$ is larger than $F(\boldsymbol{y})$ by at least a threshold. By $\theta$-domination, solutions with close $F$ values will be kept

---

**Algorithm 3** PONSS Algorithm

---

**Input**: all items $V = \{v_1, \ldots, v_n\}$, a noisy objective function $F$, and a budget $k$
**Parameter**: $T$, $\theta$ and $B$
**Output**: a subset of $V$ with at most $k$ items
**Process**:

1: Let $\boldsymbol{x} = \{0\}^n$, $P = \{\boldsymbol{x}\}$, and let $t = 0$.
2: **while** $t < T$ **do**
3:  Select $\boldsymbol{x}$ from $P$ uniformly at random.
4:  Generate $\boldsymbol{x}'$ by flipping each bit of $\boldsymbol{x}$ with probability $\frac{1}{n}$.
5:  **if** $\nexists \boldsymbol{z} \in P$ such that $\boldsymbol{z} \succ_\theta \boldsymbol{x}'$ **then**
6:    $P = (P \setminus \{\boldsymbol{z} \in P \mid \boldsymbol{x}' \succeq_\theta \boldsymbol{z}\}) \cup \{\boldsymbol{x}'\}$.
7:    $Q = \{\boldsymbol{z} \in P \mid |\boldsymbol{z}| = |\boldsymbol{x}'|\}$.
8:    **if** $|Q| = B + 1$ **then**
9:      $P = P \setminus Q$ and let $j = 0$.
10:      **while** $j < B$ **do**
11:        Select two solutions $\boldsymbol{z_1}, \boldsymbol{z_2}$ from $Q$ uniformly at random without replacement.
12:        Evaluate $F(\boldsymbol{z_1}), F(\boldsymbol{z_2})$; let $\hat{\boldsymbol{z}} = \arg\max_{\boldsymbol{z} \in \{\boldsymbol{z_1}, \boldsymbol{z_2}\}} F(\boldsymbol{z})$ (breaking ties randomly).
13:        $P = P \cup \{\hat{\boldsymbol{z}}\}$, $Q = Q \setminus \{\hat{\boldsymbol{z}}\}$, and $j = j + 1$.
14:      **end while**
15:    **end if**
16:  **end if**
17:  $t = t + 1$.
18: **end while**
19: **return** $\arg\max_{\boldsymbol{x} \in P, |\boldsymbol{x}| \leq k} F(\boldsymbol{x})$

---

in $P$ rather than only one with the best $F$ value is kept; thus the risk of removing a good solution is reduced. This modified algorithm called PONSS (Pareto Optimization for Noisy Subset Selection) is presented in Algorithm 3. However, using $\theta$-domination may also make the size of $P$ very large, and then reduce the efficiency. We further introduce a parameter $B$ to limit the number of solutions in $P$ for each possible subset size. That is, if the number of solutions with the same size in $P$ exceeds $B$, one of them will be deleted. As shown in lines 7-15, the better one of two solutions randomly selected from $Q$ is kept; this process is repeated for $B$ times, and the remaining solution in $Q$ is deleted.

For the analysis of PONSS, we consider random noise, i.e., $F(\boldsymbol{x})$ is a random variable, and assume that the probability of $F(\boldsymbol{x}) > F(\boldsymbol{y})$ is not less than $0.5 + \delta$ if $f(\boldsymbol{x}) > f(\boldsymbol{y})$, i.e.,

$$\Pr(F(\boldsymbol{x}) > F(\boldsymbol{y})) \geq 0.5 + \delta \quad \text{if} \quad f(\boldsymbol{x}) > f(\boldsymbol{y}), \tag{5}$$

where $\delta \in [0, 0.5)$. This assumption is satisfied in many noisy settings, e.g., the noise distribution is i.i.d. for each $\boldsymbol{x}$ (which is explained in the supplementary material). Note that for comparing two solutions selected from $Q$ in line 12 of PONSS, we reevaluate their noisy objective $F$ values independently, i.e., each evaluation is a new independent random draw from the noise distribution.

For the multiplicative noise model, we use the multiplicative $\theta$-domination relation as presented in Definition 5. That is, $\boldsymbol{x} \succeq_\theta \boldsymbol{y}$ if $F(\boldsymbol{x}) \geq \frac{1+\theta}{1-\theta} \cdot F(\boldsymbol{y})$ and $|\boldsymbol{x}| \leq |\boldsymbol{y}|$. The approximation ratio of PONSS with the assumption Eq. (5) is shown in Theorem 5, which is better than that of POSS under general multiplicative noise (i.e., Theorem 3), because

$$\frac{1 - \left(\frac{1-\epsilon}{1+\epsilon}\right)^k \left(1 - \frac{\gamma_{\min}}{k}\right)^k}{1 - \frac{1-\epsilon}{1+\epsilon}\left(1 - \frac{\gamma_{\min}}{k}\right)} = \sum_{i=0}^{k-1} \left(\frac{1-\epsilon}{1+\epsilon}\left(1 - \frac{\gamma_{\min}}{k}\right)\right)^i \leq \sum_{i=0}^{k-1} \left(1 - \frac{\gamma_{\min}}{k}\right)^i = \frac{1 - \left(1 - \frac{\gamma_{\min}}{k}\right)^k}{\frac{\gamma_{\min}}{k}}.$$

Particularly for the submodular case where $\gamma_{\min} = 1$, PONSS with the assumption Eq. (5) can achieve a constant approximation ratio even when $\epsilon$ is a constant, while the greedy algorithm and POSS under general multiplicative noise only guarantee a $\Theta(1/k)$ approximation ratio. Note that when $\delta$ is a constant, the approximation guarantee of PONSS can hold with a constant probability by using a polynomially large $B$, and thus the number of iterations of PONSS is polynomial in expectation.

**Definition 5** (Multiplicative $\theta$-Domination). *For two solutions $\boldsymbol{x}$ and $\boldsymbol{y}$,*

- *$\boldsymbol{x}$ weakly dominates $\boldsymbol{y}$ (denoted as $\boldsymbol{x} \succeq_\theta \boldsymbol{y}$) if $f_1(\boldsymbol{x}) \geq \frac{1+\theta}{1-\theta} \cdot f_1(\boldsymbol{y}) \wedge f_2(\boldsymbol{x}) \geq f_2(\boldsymbol{y})$;*

- *$\boldsymbol{x}$ dominates $\boldsymbol{y}$ (denoted as $\boldsymbol{x} \succ_\theta \boldsymbol{y}$) if $\boldsymbol{x} \succeq_\theta \boldsymbol{y}$ and $f_1(\boldsymbol{x}) > \frac{1+\theta}{1-\theta} \cdot f_1(\boldsymbol{y}) \vee f_2(\boldsymbol{x}) > f_2(\boldsymbol{y})$.*

**Lemma 1.** *[21] For any $X \subseteq V$, there exists one item $\hat{v} \in V \setminus X$ such that*

$$f(X \cup \{\hat{v}\}) - f(X) \geq \frac{\gamma_{X,k}}{k}(OPT - f(X)).$$

**Theorem 5.** *For subset selection under multiplicative noise with the assumption Eq. (5), with probability at least $\frac{1}{2}(1 - \frac{12nk^2 \log 2k}{B^2 \delta})$, PONSS using $\theta \geq \epsilon$ and $T = 2eBnk^2 \log 2k$ finds a subset $X$ with $|X| \leq k$ and*

$$f(X) \geq \frac{1-\epsilon}{1+\epsilon}\left(1 - \left(1 - \frac{\gamma_{\min}}{k}\right)^k\right) \cdot OPT.$$

*Proof.* Let $J_{\max}$ denote the maximum value of $j \in [0, k]$ such that in $P$, there exists a solution $x$ with $|x| \leq j$ and $f(x) \geq (1 - (1 - \frac{\gamma_{\min}}{k})^j) \cdot OPT$. Note that $J_{\max} = k$ implies that there exists one solution $x^*$ in $P$ satisfying that $|x^*| \leq k$ and $f(x^*) \geq (1 - (1 - \frac{\gamma_{\min}}{k})^k) \cdot OPT$. Since the final selected solution $x$ from $P$ has the largest $F$ value (i.e., line 19 of Algorithm 3), we have

$$f(x) \geq \frac{1}{1+\epsilon}F(x) \geq \frac{1}{1+\epsilon}F(x^*) \geq \frac{1-\epsilon}{1+\epsilon}f(x^*).$$

That is, the desired approximation bound is reached. Thus, we only need to analyze the probability of $J_{\max} = k$ after running $T = 2eBnk^2 \log 2k$ number of iterations.

Assume that in the run of PONSS, one solution with the best $f$ value in $Q$ is always kept after each implementation of lines 8-15. We then show that $J_{\max}$ can reach $k$ with probability at least 0.5 after $2eBnk^2 \log 2k$ iterations. $J_{\max}$ is initially 0 since it starts from $\{0\}^n$, and we assume that currently $J_{\max} = i < k$. Let $x$ be a corresponding solution with the value $i$, i.e., $|x| \leq i$ and

$$f(x) \geq \left(1 - \left(1 - \frac{\gamma_{\min}}{k}\right)^i\right) \cdot OPT. \tag{6}$$

First, $J_{\max}$ will not decrease. If $x$ is not deleted, it obviously holds. For deleting $x$, there are two possible cases. If $x$ is deleted in line 6, the newly included solution $x' \succeq_\theta x$, which implies that $|x'| \leq |x| \leq i$ and $f(x') \geq \frac{1}{1+\epsilon}F(x') \geq \frac{1}{1+\epsilon} \cdot \frac{1+\theta}{1-\theta}F(x) \geq \frac{1}{1+\epsilon} \cdot \frac{1+\epsilon}{1-\epsilon}F(x) \geq f(x)$, where the third inequality is by $\theta \geq \epsilon$. If $x$ is deleted in lines 8-15, there must exist one solution $z^*$ in $P$ with $|z^*| = |x|$ and $f(z^*) \geq f(x)$, because we assume that one solution with the best $f$ value in $Q$ is kept. Second, $J_{\max}$ can increase in each iteration with some probability. From Lemma 1, we know that a new solution $x'$ can be produced by flipping one specific 0 bit of $x$ (i.e., adding a specific item) such that $|x'| = |x| + 1 \leq i + 1$ and

$$f(x') \geq \left(1 - \frac{\gamma_{x,k}}{k}\right)f(x) + \frac{\gamma_{x,k}}{k} \cdot OPT \geq \left(1 - \left(1 - \frac{\gamma_{\min}}{k}\right)^{i+1}\right) \cdot OPT,$$

where the second inequality is by Eq. (6) and $\gamma_{x,k} \geq \gamma_{\min}$ (since $|x| < k$ and $\gamma_{x,k}$ decreases with $x$). Note that $x'$ will be added into $P$; otherwise, there must exist one solution in $P$ dominating $x'$ (line 5 of Algorithm 3), and this implies that $J_{\max}$ has already been larger than $i$, which contradicts with the assumption $J_{\max} = i$. After including $x'$, $J_{\max} \geq i + 1$. Since $P$ contains at most $B$ solutions for each possible size $\{0, \ldots, 2k-1\}$, $|P| \leq 2Bk$. Thus, $J_{\max}$ can increase by at least 1 in one iteration with probability at least $\frac{1}{|P|} \cdot \frac{1}{n}(1 - \frac{1}{n})^{n-1} \geq \frac{1}{2eBnk}$, where $\frac{1}{|P|}$ is the probability of selecting $x$ in line 3 of Algorithm 3 due to uniform selection and $\frac{1}{n}(1 - \frac{1}{n})^{n-1}$ is the probability of flipping only a specific bit of $x$ in line 4. We divide the $2eBnk^2 \log 2k$ iterations into $k$ phases with equal length. For reaching $J_{\max} = k$, it is sufficient that $J_{\max}$ increases at least once in each phase. Thus, we have

$$\Pr(J_{\max} = k) \geq \left(1 - (1 - 1/(2eBnk))^{2eBnk \log 2k}\right)^k \geq (1 - 1/(2k))^k \geq 1/2.$$

We then only need to investigate our assumption that in the run of $2eBnk^2 \log 2k$ iterations, when implementing lines 8-15, one solution with the best $f$ value in $Q$ is always kept. Let $R = \{z^* \in \arg\max_{z \in Q} f(z)\}$. If $|R| > 1$, it trivially holds, since only one solution from $Q$ is deleted. If $|R| = 1$, deleting the solution $z^*$ with the best $f$ value implies that $z^*$ is never included into $P$ in implementing lines 11-13 of Algorithm 3, which are repeated for $B$ iterations. In the $j$-th (where $0 \leq j \leq B - 1$) iteration, $|Q| = B + 1 - j$. Under the condition that $z^*$ is not included into $P$ from the 0-th to the $(j-1)$-th iteration, the probability that $z^*$ is selected in line 11 is $(B - j)/\binom{B+1-j}{2} = 2/(B + 1 - j)$. We know from Eq. (5) that $F(z^*)$ is better in the comparison of line 12 with probability at least $0.5 + \delta$. Thus, the probability of not including $z^*$ into $P$ in the $j$-th

iteration is at most $1-\frac{2}{B+1-j}\cdot(0.5+\delta)$. Then, the probability of deleting the solution with the best $f$ value in $Q$ when implementing lines 8-15 is at most $\prod_{j=0}^{B-1}(1-\frac{1+2\delta}{B+1-j})$. Taking the logarithm, we get

$$\sum_{j=0}^{B-1}\log\left(\frac{B-j-2\delta}{B+1-j}\right)=\sum_{j=1}^{B}\log\left(\frac{j-2\delta}{j+1}\right)\leq\int_{1}^{B+1}\log\left(\frac{j-2\delta}{j+1}\right)\mathrm{d}j$$

$$=\log\left(\frac{(B+1-2\delta)^{B+1-2\delta}}{(B+2)^{B+2}}\right)-\log\left(\frac{(1-2\delta)^{1-2\delta}}{2^2}\right),$$

where the inequality is since $\log\frac{j-2\delta}{j+1}$ is increasing with $j$, and the last equality is since the derivative of $\log\frac{(j-2\delta)^{j-2\delta}}{(j+1)^{j+1}}$ with respect to $j$ is $\log\frac{j-2\delta}{j+1}$. Thus, we have

$$\prod_{j=0}^{B-1}\left(1-\frac{1+2\delta}{B+1-j}\right)\leq\left(\frac{B+1-2\delta}{B+2}\right)^{B+2}\cdot\frac{1}{(B+1-2\delta)^{1+2\delta}}\cdot\frac{4}{(1-2\delta)^{1-2\delta}}\leq\frac{4}{e^{1-1/e}B^{1+2\delta}},$$

where the last inequality is by $0<1-2\delta\leq1$ and $(1-2\delta)^{1-2\delta}\geq e^{-1/e}$. By the union bound, our assumption holds with probability at least $1-(12nk^2\log 2k)/B^{2\delta}$. Thus, the theorem holds. $\qquad\square$

For the additive noise model, we use the additive $\theta$-domination relation as presented in Definition 6. That is, $\boldsymbol{x}\succeq_\theta\boldsymbol{y}$ if $F(\boldsymbol{x})\geq F(\boldsymbol{y})+2\theta$ and $|\boldsymbol{x}|\leq|\boldsymbol{y}|$. By applying Eq. (3) and additive $\theta$-domination to the proof procedure of Theorem 5, we can prove the approximation ratio of PONSS under additive noise with the assumption Eq. (5), as shown in Theorem 6. Compared with the approximation ratio of POSS under general additive noise (i.e., Theorem 4), PONSS achieves a better one. This can be easily verified since $(1-(1-\frac{\gamma_{\min}}{k})^k)\frac{2k\epsilon}{\gamma_{\min}}\geq 2\epsilon$, where the inequality is derived by $\gamma_{\min}\in[0,1]$.

**Definition 6** (Additive $\theta$-Domination). *For two solutions $\boldsymbol{x}$ and $\boldsymbol{y}$,*
- *$\boldsymbol{x}$ weakly dominates $\boldsymbol{y}$ (denoted as $\boldsymbol{x}\succeq_\theta\boldsymbol{y}$) if $f_1(\boldsymbol{x})\geq f_1(\boldsymbol{y})+2\theta\wedge f_2(\boldsymbol{x})\geq f_2(\boldsymbol{y})$;*
- *$\boldsymbol{x}$ dominates $\boldsymbol{y}$ (denoted as $\boldsymbol{x}\succ_\theta\boldsymbol{y}$) if $\boldsymbol{x}\succeq_\theta\boldsymbol{y}$ and $f_1(\boldsymbol{x})>f_1(\boldsymbol{y})+2\theta\vee f_2(\boldsymbol{x})>f_2(\boldsymbol{y})$.*

**Theorem 6.** *For subset selection under additive noise with the assumption Eq. (5), with probability at least $\frac{1}{2}(1-\frac{12nk^2\log 2k}{B^{2\delta}})$, PONSS using $\theta\geq\epsilon$ and $T=2eBnk^2\log 2k$ finds a subset $X$ with $|X|\leq k$ and*

$$f(X)\geq\left(1-\left(1-\frac{\gamma_{\min}}{k}\right)^k\right)\cdot OPT-2\epsilon.$$

## 6 Empirical Study

We conducted experiments on two typical subset selection problems: influence maximization and sparse regression, where the former has a submodular objective function and the latter has a non-submodular one. The number $T$ of iterations in POSS is set to $2ek^2n$ as suggested by Theorem 3. For PONSS, $B$ is set to $k$, and $\theta$ is set to 1, which is obviously not smaller than $\epsilon$. Note that POSS needs one objective evaluation for the newly generated solution $\boldsymbol{x}'$ in each iteration, while PONSS needs 1 or $1+2B$ evaluations, which depends on whether the condition in line 8 of Algorithm 3 is satisfied. For the fairness of comparison, PONSS is terminated until the total number of evaluations reaches that of POSS, i.e., $2ek^2n$. Note that in the run of each algorithm, only a noisy objective value $F$ can be obtained; while for the final output solution, we report its accurately estimated $f$ value for the assessment of the algorithms by an expensive evaluation. As POSS and PONSS are randomized algorithms and the behavior of the greedy algorithm is also randomized under random noise, we repeat the run 10 times independently and report the average estimated $f$ values.

**Influence Maximization** The task is to identify a set of influential users in social networks. Let a directed graph $G(V,E)$ represent a social network, where each node is a user and each edge $(u,v)\in E$ has a probability $p_{u,v}$ representing the influence strength from user $u$ to $v$. Given a budget $k$, influence maximization is to find a subset $X$ of $V$ with $|X|\leq k$ such that the expected number of nodes activated by propagating from $X$ (called influence spread) is maximized. The fundamental propagation model Independent Cascade [11] is used. Note that the set of active nodes in the diffusion process is a random variable, and the expectation of its size is monotone and submodular [16].

We use two real-world data sets: *ego-Facebook* and *Weibo*. *ego-Facebook* is downloaded from `http://snap.stanford.edu/data/index.html`, and *Weibo* is crawled from a Chinese microblogging

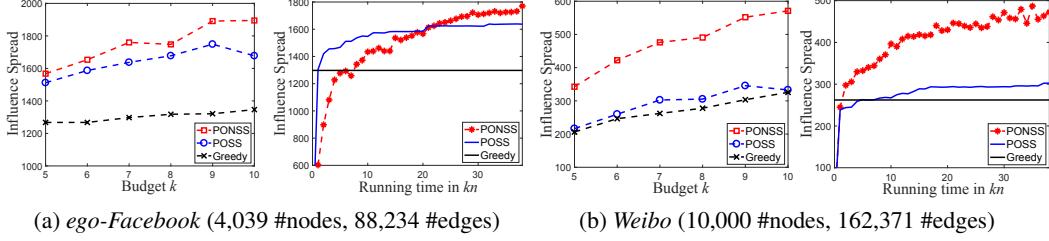

(a) *ego-Facebook* (4,039 #nodes, 88,234 #edges)  (b) *Weibo* (10,000 #nodes, 162,371 #edges)

Figure 2: Influence maximization (influence spread: the larger the better). The right subfigure on each data set: influence spread vs running time of PONSS and POSS for $k = 7$.

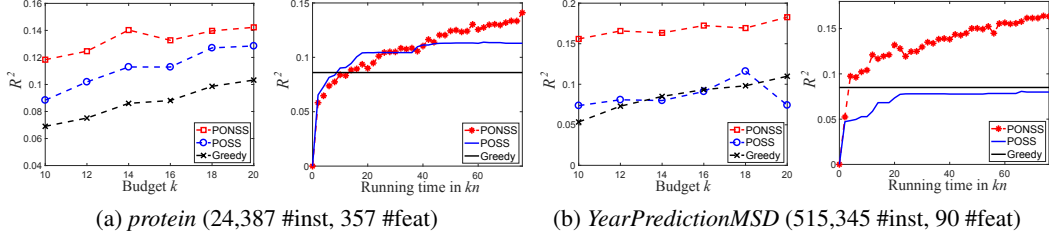

(a) *protein* (24,387 #inst, 357 #feat)  (b) *YearPredictionMSD* (515,345 #inst, 90 #feat)

Figure 3: Sparse regression ($R^2$: the larger the better). The right subfigure on each data set: $R^2$ vs running time of PONSS and POSS for $k = 14$.

site Weibo.com like Twitter. On each network, the propagation probability of one edge from node $u$ to $v$ is estimated by $\frac{weight(u,v)}{indegree(v)}$, as widely used in [3, 12]. We test the budget $k$ from 5 to 10. For estimating the objective influence spread, we simulate the diffusion process 10 times independently and use the average as an estimation. But for the final output solutions of the algorithms, we average over 10,000 times for accurate estimation.

From the left subfigure on each data set in Figure 2, we can see that POSS is better than the greedy algorithm, and PONSS performs the best. By selecting the greedy algorithm as the baseline, we plot in the right subfigures the curve of influence spread over running time for PONSS and POSS with $k = 7$. Note that the $x$-axis is in $kn$, the running time order of the greedy algorithm. We can see that PONSS quickly reaches a better performance, which implies that PONSS can be efficient in practice.

**Sparse Regression** The task is to find a sparse approximation solution to the linear regression problem. Given all observation variables $V = \{v_1, \ldots, v_n\}$, a predictor variable $z$ and a budget $k$, sparse regression is to find a set of at most $k$ variables maximizing the *squared multiple correlation* $R^2_{z,X} = 1 - \mathrm{MSE}_{z,X}$ [8, 15], where $\mathrm{MSE}_{z,X} = \min_{\boldsymbol{\alpha} \in \mathbb{R}^{|X|}} \mathbb{E}[(z - \sum_{i \in X} \alpha_i v_i)^2]$ denotes the *mean squared error*. We assume w.l.o.g. that all random variables are normalized to have expectation 0 and variance 1. The objective $R^2_{z,X}$ is monotone increasing, but not necessarily submodular [6].

We use two data sets from `http://www.csie.ntu.edu.tw/~cjlin/libsvmtools/datasets/`. The budget $k$ is set to $\{10, 12, \ldots, 20\}$. For estimating $R^2$ in the optimization process, we use a random sample of 1000 instances. But for the final output solutions, we use the whole data set for accurate estimation. The results are plotted in Figure 3. The performances of the three algorithms are similar to that observed for influence maximization, except some losses of POSS over the greedy algorithm (e.g., on *YearPredictionMSD* with $k = 20$).

For both tasks, we test PONSS with $\theta = \{0.1, 0.2, \ldots, 1\}$. The results are provided in the supplementary material due to space limitation, which show that PONSS is always better than POSS and the greedy algorithm. This implies that the performance of PONSS is not sensitive to the value of $\theta$.

# 7   Conclusion

In this paper, we study the subset selection problem with monotone objective functions under multiplicative and additive noises. We first show that the greedy algorithm and POSS, two powerful algorithms for noise-free subset selection, achieve nearly the same approximation guarantee under noise. Then, we propose a new algorithm PONSS, which can achieve a better approximation ratio with some assumption such as i.i.d. noise distribution. The experimental results on influence maximization and sparse regression exhibit the superior performance of PONSS.

**Acknowledgements**   The authors would like to thank reviewers for their helpful comments and suggestions. C. Qian was supported by NSFC (61603367) and YESS (2016QNRC001). Y. Yu was supported by JiangsuSF (BK20160066, BK20170013). K. Tang was supported by NSFC (61672478) and Royal Society Newton Advanced Fellowship (NA150123). Z.-H. Zhou was supported by NSFC (61333014) and Collaborative Innovation Center of Novel Software Technology and Industrialization.

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
