[Supplementary Material]

# Supplementary Material: Subset Selection under Noise

**Chao Qian**[1] **Jing-Cheng Shi**[2] **Yang Yu**[2] **Ke Tang**[3,1] **Zhi-Hua Zhou**[2]

[1]Anhui Province Key Lab of Big Data Analysis and Application, USTC, China
[2]National Key Lab for Novel Software Technology, Nanjing University, China
[3]Shenzhen Key Lab of Computational Intelligence, SUSTech, China

chaoqian@ustc.edu.cn    tangk3@sustc.edu.cn
{shijc,yuy,zhouzh}@lamda.nju.edu.cn

## 1 Detailed Proofs

This part aims to provide some detailed proofs, which are omitted in our original paper due to space limitation.

**Proof of Theorem 1.** Let $X^*$ be an optimal subset, i.e., $f(X^*) = OPT$. Let $X_i$ denote the subset after the $i$-th iteration of the greedy algorithm. Then, we have

$$f(X^*) - f(X_i) \leq f(X^* \cup X_i) - f(X_i)$$

$$\leq \frac{1}{\gamma_{X_i,k}} \sum_{v \in X^* \setminus X_i} \big(f(X_i \cup \{v\}) - f(X_i)\big)$$

$$\leq \frac{1}{\gamma_{X_i,k}} \sum_{v \in X^* \setminus X_i} \left(\frac{1}{1-\epsilon} F(X_i \cup \{v\}) - f(X_i)\right)$$

$$\leq \frac{1}{\gamma_{X_i,k}} \sum_{v \in X^* \setminus X_i} \left(\frac{1}{1-\epsilon} F(X_{i+1}) - f(X_i)\right)$$

$$\leq \frac{k}{\gamma_{X_k,k}} \left(\frac{1+\epsilon}{1-\epsilon} f(X_{i+1}) - f(X_i)\right),$$

where the first inequality is by the monotonicity of $f$, the second inequality is by the definition of submodularity ratio and $|X^*| \leq k$, the third is by the definition of multiplicative noise, i.e., $F(X) \geq (1-\epsilon) \cdot f(X)$, the fourth is by line 3 of Algorithm 1, and the last is by $\gamma_{X_i,k} \geq \gamma_{X_{i+1},k}$ and $F(X) \leq (1+\epsilon) \cdot f(X)$. By a simple transformation, we can equivalently get

$$f(X_{i+1}) \geq \left(\frac{1-\epsilon}{1+\epsilon}\right) \left(\left(1 - \frac{\gamma_{X_k,k}}{k}\right) f(X_i) + \frac{\gamma_{X_k,k}}{k} OPT\right).$$

Based on this inequality, an inductive proof gives the approximation ratio of the returned subset $X_k$:

$$f(X_k) \geq \frac{\frac{1-\epsilon}{1+\epsilon} \frac{\gamma_{X_k,k}}{k}}{1 - \frac{1-\epsilon}{1+\epsilon}\left(1 - \frac{\gamma_{X_k,k}}{k}\right)} \left(1 - \left(\frac{1-\epsilon}{1+\epsilon}\right)^k \left(1 - \frac{\gamma_{X_k,k}}{k}\right)^k\right) \cdot OPT.$$

$\square$

Lemma 2 shows the relation between the $F$ values of adjacent subsets, which will be used in the proof of Theorem 3.

**Lemma 2.** *For any $X \subseteq V$, there exists one item $\hat{v} \in V \setminus X$ such that*

$$F(X \cup \{\hat{v}\}) \geq \left(\frac{1-\epsilon}{1+\epsilon}\right)\left(1 - \frac{\gamma_{X,k}}{k}\right) F(X) + \frac{(1-\epsilon)\gamma_{X,k}}{k} \cdot OPT.$$

*Proof.* Let $X^*$ be an optimal subset, i.e., $f(X^*) = OPT$. Let $\hat{v} \in \arg\max_{v \in X^* \setminus X} F(X \cup \{v\})$. Then, we have

$$f(X^*) - f(X) \leq f(X^* \cup X) - f(X)$$
$$\leq \frac{1}{\gamma_{X,k}} \sum_{v \in X^* \setminus X} \left(f(X \cup \{v\}) - f(X)\right)$$
$$\leq \frac{1}{\gamma_{X,k}} \sum_{v \in X^* \setminus X} \left(\frac{1}{1-\epsilon} F(X \cup \{v\}) - f(X)\right)$$
$$\leq \frac{k}{\gamma_{X,k}} \left(\frac{1}{1-\epsilon} F(X \cup \{\hat{v}\}) - f(X)\right),$$

where the first inequality is by the monotonicity of $f$, the second inequality is by the definition of submodularity ratio and $|X^*| \leq k$, and the third is by $F(X) \geq (1 - \epsilon)f(X)$. By a simple transformation, we can equivalently get

$$F(X \cup \{\hat{v}\}) \geq (1 - \epsilon)\left(\left(1 - \frac{\gamma_{X,k}}{k}\right) f(X) + \frac{\gamma_{X,k}}{k} \cdot OPT\right).$$

By applying $f(X) \geq F(X)/(1+\epsilon)$ to this inequality, the lemma holds. $\qquad \square$

**Proof of Theorem 3.** Let $J_{\max}$ denote the maximum value of $j \in [0, k]$ such that in $P$, there exists a solution $\boldsymbol{x}$ with $|\boldsymbol{x}| \leq j$ and

$$F(\boldsymbol{x}) \geq \frac{(1-\epsilon)\frac{\gamma_{\min}}{k}}{1 - \frac{1-\epsilon}{1+\epsilon}\left(1 - \frac{\gamma_{\min}}{k}\right)} \left(1 - \left(\frac{1-\epsilon}{1+\epsilon}\right)^j \left(1 - \frac{\gamma_{\min}}{k}\right)^j\right) \cdot OPT.$$

We analyze the expected number of iterations until $J_{\max} = k$, which implies that there exists one solution $\boldsymbol{x}$ in $P$ satisfying that $|\boldsymbol{x}| \leq k$ and $F(\boldsymbol{x}) \geq \frac{(1-\epsilon)\frac{\gamma_{\min}}{k}}{1 - \frac{1-\epsilon}{1+\epsilon}\left(1 - \frac{\gamma_{\min}}{k}\right)}(1 - (\frac{1-\epsilon}{1+\epsilon})^k(1 - \frac{\gamma_{\min}}{k})^k) \cdot OPT$. Since $f(\boldsymbol{x}) \geq F(\boldsymbol{x})/(1+\epsilon)$, the desired approximation bound has been reached when $J_{\max} = k$.

The initial value of $J_{\max}$ is 0, since POSS starts from $\{0\}^n$. Assume that currently $J_{\max} = i < k$. Let $\boldsymbol{x}$ be a corresponding solution with the value $i$, i.e., $|\boldsymbol{x}| \leq i$ and

$$F(\boldsymbol{x}) \geq \frac{(1-\epsilon)\frac{\gamma_{\min}}{k}}{1 - \frac{1-\epsilon}{1+\epsilon}\left(1 - \frac{\gamma_{\min}}{k}\right)} \left(1 - \left(\frac{1-\epsilon}{1+\epsilon}\right)^i \left(1 - \frac{\gamma_{\min}}{k}\right)^i\right) \cdot OPT. \qquad (1)$$

It is easy to see that $J_{\max}$ cannot decrease because deleting $\boldsymbol{x}$ from $P$ (line 6 of Algorithm 2) implies that $\boldsymbol{x}$ is weakly dominated by the newly generated solution $\boldsymbol{x}'$, which must have a smaller size and a larger $F$ value. By Lemma 2, we know that flipping one specific 0 bit of $\boldsymbol{x}$ (i.e., adding a specific item) can generate a new solution $\boldsymbol{x}'$, which satisfies that

$$F(\boldsymbol{x}') \geq \left(\frac{1-\epsilon}{1+\epsilon}\right)\left(1 - \frac{\gamma_{\boldsymbol{x},k}}{k}\right) F(\boldsymbol{x}) + \frac{(1-\epsilon)\gamma_{\boldsymbol{x},k}}{k} \cdot OPT$$
$$= \frac{1-\epsilon}{1+\epsilon} F(\boldsymbol{x}) + \left(OPT - \frac{F(\boldsymbol{x})}{1+\epsilon}\right)\frac{(1-\epsilon)\gamma_{\boldsymbol{x},k}}{k}.$$

Note that $OPT - \frac{F(\boldsymbol{x})}{1+\epsilon} \geq f(\boldsymbol{x}) - \frac{F(\boldsymbol{x})}{1+\epsilon} \geq 0$. Moreover, $\gamma_{\boldsymbol{x},k} \geq \gamma_{\min}$, since $|\boldsymbol{x}| < k$ and $\gamma_{\boldsymbol{x},k}$ decreases with $\boldsymbol{x}$. Thus, we have

$$F(\boldsymbol{x}') \geq \left(\frac{1-\epsilon}{1+\epsilon}\right)\left(1 - \frac{\gamma_{\min}}{k}\right) F(\boldsymbol{x}) + \frac{(1-\epsilon)\gamma_{\min}}{k} \cdot OPT.$$

By applying Eq. (1) to the above inequality, we easily get

$$F(\boldsymbol{x}') \geq \frac{(1-\epsilon)\frac{\gamma_{\min}}{k}}{1 - \frac{1-\epsilon}{1+\epsilon}\left(1 - \frac{\gamma_{\min}}{k}\right)} \left(1 - \left(\frac{1-\epsilon}{1+\epsilon}\right)^{i+1} \left(1 - \frac{\gamma_{\min}}{k}\right)^{i+1}\right) \cdot OPT.$$

Since $|\boldsymbol{x}'| = |\boldsymbol{x}| + 1 \leq i + 1$, $\boldsymbol{x}'$ will be included into $P$; otherwise, $\boldsymbol{x}'$ must be dominated by one solution in $P$ (line 5 of Algorithm 2), and this implies that $J_{\max}$ has already been larger than $i$, which contradicts with the assumption $J_{\max} = i$. After including $\boldsymbol{x}'$, $J_{\max} \geq i + 1$. Let $P_{\max}$ denote the largest size of $P$ during the run of POSS. Thus, $J_{\max}$ can increase by at least 1 in one iteration with probability at least $\frac{1}{P_{\max}} \cdot \frac{1}{n}(1 - \frac{1}{n})^{n-1} \geq \frac{1}{enP_{\max}}$, where $\frac{1}{P_{\max}}$ is a lower bound on the probability of selecting $\boldsymbol{x}$ in line 3 of Algorithm 2 and $\frac{1}{n}(1 - \frac{1}{n})^{n-1}$ is the probability of flipping only a specific bit of $\boldsymbol{x}$ in line 4. Then, it needs at most $enP_{\max}$ expected number of iterations to increase $J_{\max}$. Thus, after $k \cdot enP_{\max}$ expected number of iterations, $J_{\max}$ must have reached $k$.

From the procedure of POSS, we know that the solutions in $P$ must be non-dominated. Thus, each value of one objective can correspond to at most one solution in $P$. Because the solutions with $|\boldsymbol{x}| \geq 2k$ have $-\infty$ value on the first objective, they must be excluded from $P$. Thus, $P_{\max} \leq 2k$, which implies that the expected number of iterations $\mathbb{E}[T]$ for finding the desired solution is at most $2ek^2n$. $\qquad\qquad\qquad\qquad\qquad\qquad\qquad\qquad\qquad\qquad\qquad\qquad\qquad\qquad\qquad\quad$ □

**Proof of Proposition 1.** Let $\mathcal{A} = \{S_1, \ldots, S_l\}$ and $\mathcal{B} = \{S_{l+1}, \ldots, S_{2l}\}$. For the greedy algorithm, if without noise, it will first select one $S_i$ from $\mathcal{A}$, and continue to select $S_i$ from $\mathcal{B}$ until reaching the budget. Thus, the greedy algorithm can find an optimal solution. But in the presence of noise, after selecting one $S_i$ from $\mathcal{A}$, it will continue to select $S_i$ from $\mathcal{A}$ rather than from $\mathcal{B}$, since for all $X \subseteq \mathcal{A}, S_i \in \mathcal{B}$, $F(X) = 2 + \delta > 2 = F(X \cup \{S_i\})$. The approximation ratio thus is only $2/(k+1)$.

For POSS under noise, we show that it can efficiently follow the path $\{0\}^n$ (i.e., $\emptyset$) $\rightarrow \{S\} \rightarrow \{S\} \cup X_2 \rightarrow \{S\} \cup X_3 \rightarrow \cdots \rightarrow \{S\} \cup X_{k-1}$ (i.e., an optimal solution), where $S$ denotes any element from $\mathcal{A}$ and $X_i$ denotes any subset of $\mathcal{B}$ with size $i$. Note that the solutions on the path will always be kept in the archive $P$ once found, because there is no other solution which can dominate them. The probability of the first "$\rightarrow$" on the path is at least $\frac{1}{P_{\max}} \cdot \frac{l}{n}(1 - \frac{1}{n})^{n-1}$, since it is sufficient to select $\{0\}^n$ in line 3 of Algorithm 2, and flip one of its first $l$ 0-bits and keep other bits unchanged in line 4. **[Multi-bit search]** *For the second "$\rightarrow$", the probability is at least $\frac{1}{P_{\max}} \cdot \frac{\binom{l}{2}}{n^2}(1 - \frac{1}{n})^{n-2}$, since it is sufficient to select $\{S\}$ and flip any two 0-bits in its second half.* For the $i$-th "$\rightarrow$" with $3 \leq i \leq k - 1$, the probability is at least $\frac{1}{P_{\max}} \cdot \frac{l-i+1}{n}(1 - \frac{1}{n})^{n-1}$, since it is sufficient to select the left solution of "$\rightarrow$" and flip one 0-bit in its second half. Thus, starting from $\{0\}^n$, POSS can follow the path in

$$enP_{\max} \cdot \left( \frac{1}{l} + \frac{4}{l-1} + \sum_{i=3}^{k-1} \frac{1}{l-i+1} \right) = O(nP_{\max} \log n)$$

expected number of iterations. Since $P_{\max} \leq 2k$, the number of iterations for finding an optimal solution is $O(kn \log n)$ in expectation. $\qquad\qquad\qquad\qquad\qquad\qquad\qquad\qquad\qquad\quad$ □

**Proof of Proposition 2.** For the greedy algorithm, if without noise, it will first select $S_{4l-2}$ since $|S_{4l-2}|$ is the largest, and then find the optimal solution $\{S_{4l-2}, S_{4l-1}\}$. But in the presence of noise, $S_{4l}$ will be first selected since $F(\{S_{4l}\}) = 2l$ is the largest, and then the solution $\{S_{4l}, S_{4l-1}\}$ is found. The approximation ratio is thus only $(3l - 2)/(4l - 3)$.

For POSS under noise, we first show that it can efficiently follow the path $\{0\}^n \rightarrow \{S_{4l}\} \rightarrow \{S_{4l}, S_{4l-1}\} \rightarrow \{S_{4l-2}, S_{4l-1}, *\}$, where $*$ denotes any subset $S_i$ with $i \neq 4l - 2, 4l - 1$. In this procedure, we can pessimistically assume that the optimal solution $\{S_{4l-2}, S_{4l-1}\}$ will never be found, since we are to derive a running time upper bound for finding it. Note that the solutions on the path will always be kept in $P$ once found, because no other solutions can dominate them. The probability of "$\rightarrow$" is at least $\frac{1}{P_{\max}} \cdot \frac{1}{n}(1 - \frac{1}{n})^{n-1} \geq \frac{1}{enP_{\max}}$, since it is sufficient to select the solution on the left of "$\rightarrow$" and flip only one specific 0-bit. Thus, starting from $\{0\}^n$, POSS can follow the path in $3 \cdot enP_{\max}$ expected number of iterations. **[Backward search]** *After that, the optimal solution $\{S_{4l-2}, S_{4l-1}\}$ can be found by selecting $\{S_{4l-2}, S_{4l-1}, *\}$ and flipping a specific 1-bit, which happens with probability at least $\frac{1}{enP_{\max}}$.* Thus, the total number of required iterations is at most $4enP_{\max}$ in expectation. Since $P_{\max} \leq 4$, $\mathbb{E}[T] = O(n)$. $\qquad\qquad\qquad$ □

For the analysis of PONSS in the original paper, we assume that
$$\Pr(F(\boldsymbol{x}) > F(\boldsymbol{y})) \geq 0.5 + \delta \quad \text{if} \quad f(\boldsymbol{x}) > f(\boldsymbol{y}),$$

where $\delta \in [0, 0.5)$. To show that this assumption holds with i.i.d. noise distribution, we prove the following claim. Note that the value of $\delta$ depends on the concrete noise distribution.

**Claim 1.** *If the noise distribution is i.i.d. for each solution $\boldsymbol{x}$, it holds that*

$$\Pr(F(\boldsymbol{x}) > F(\boldsymbol{y})) \geq 0.5 \quad if \quad f(\boldsymbol{x}) > f(\boldsymbol{y}).$$

*Proof.* If $F(\boldsymbol{x}) = f(\boldsymbol{x}) + \xi(\boldsymbol{x})$, where the noise $\xi(\boldsymbol{x})$ is drawn independently from the same distribution for each $\boldsymbol{x}$, we have, for two solutions $\boldsymbol{x}$ and $\boldsymbol{y}$ with $f(\boldsymbol{x}) > f(\boldsymbol{y})$,

$$\begin{aligned}
\Pr(F(\boldsymbol{x}) > F(\boldsymbol{y})) &= \Pr(f(\boldsymbol{x}) + \xi(\boldsymbol{x}) > f(\boldsymbol{y}) + \xi(\boldsymbol{y})) \\
&\geq \Pr(\xi(\boldsymbol{x}) \geq \xi(\boldsymbol{y})) \\
&\geq 0.5,
\end{aligned}$$

where the first inequality is by the condition that $f(\boldsymbol{x}) > f(\boldsymbol{y})$, and the last inequality is derived by $\Pr(\xi(\boldsymbol{x}) \geq \xi(\boldsymbol{y})) + \Pr(\xi(\boldsymbol{x}) \leq \xi(\boldsymbol{y})) \geq 1$ and $\Pr(\xi(\boldsymbol{x}) \geq \xi(\boldsymbol{y})) = \Pr(\xi(\boldsymbol{x}) \leq \xi(\boldsymbol{y}))$ due to that $\xi(\boldsymbol{x})$ and $\xi(\boldsymbol{y})$ are from the same distribution.

If $F(\boldsymbol{x}) = f(\boldsymbol{x}) \cdot \xi(\boldsymbol{x})$, the claim holds similarly. $\qquad\square$

## 2  Detailed Experimental Results

This part aims to provide some experimental results, which are omitted in our original paper due to space limitation.

(a) *ego-Facebook*  (b) *Weibo*

Figure 1: Influence maximization with the budget $k = 7$ (influence spread: the larger the better): the comparison between PONSS with different $\theta$ values, POSS and the greedy algorithm.

(a) *protein*  (b) *YearPredictionMSD*

Figure 2: Sparse regression with the budget $k = 14$ ($R^2$: the larger the better): the comparison between PONSS with different $\theta$ values, POSS and the greedy algorithm.