[Reviews · NeurIPS 2017]

Reviewer 1



This paper considers the problem of maximizing a monotone set function subject to a cardinality constraint. The authors consider a novel combination of functions with both bounded submodularity ratio and additive noise. These setting have been considered separately before, but a joint analysis leads to a novel algorithm PONSS. This has improved theoretical guarantees and experimental performance when compared to previous noise-agnostic greedy algorithms. The paper flows well and is generally a pleasure to read. Clarity Small typo: "better on subclass problems" on line 26 The paper could benefit from a more thorough literature survey, including more recent results on the tightness of the submodularity ratio and applications to sparse regression [1] [2]. One comment on the beginning of Section 2: in this setting minimizing f subject to a cardinality constraint is not the same as maximizing -f subject to a cardinality constraint. Generally if -f is (weakly) submodular then f is (weakly) supermodular, and -f will not have the same submodularity ratio as f. I would suggest including "i.i.d. noise distribution" when the assumptions for PONSS instead are first mentioned on page 2 ("fixed noise distribution"), instead of waiting until Section 5. Impact Influence maximization is an interesting application of noisy submodular optimization, since the objective function is often hard to compute exactly. The authors should address that the PONSS guarantees hold with probability < 1/2 while the other guarantees are deterministic, and explain whether this affects the claim that the PONSS approximation ratio is "better". O(Bnk^2) iterations can be prohibitive for many applications, and approximate/accelerated versions of the standard greedy algorithm can run in sublinear time. Experiments Most of my concerns are with the experiments section. In the sparse regression experiment, the authors report final performance on the entire dataset. They should explain why the did not optimize on a training set and then report final performance on a test set. Typically the training set is used to check how algorithms perform on the optimization, and the test set is used to check whether the solution generalizes without overfitting. Additionally, it appears that all of the R^2 values continue to increase at the largest value k. The authors should continue running for larger k or state that they were limited by large running times of the algorithms. It would be more interesting to compare PONSS with other greedy algorithms (double greedy, random greedy, forward backward greedy, etc.), which are known to perform better than standard greedy on general matroid constraints Questions -What is the meaning of multiplicative domination in PONSS experiments if the noise estimate parameter \theta is set to 1? It would be interesting to quantify the drop in performance when 1 > \theta^' > \epsilon is used instead of \epsilon. -For what range of memory B do the claims on line 170 hold? -Could PONSS be parallellized or otherwise accelerated to reduce running time? [1] A. A. Bian, J. M. Buhmann, A. Krause, and S. Tschiatschek. Guarantees for greedy maximization of non-submodular functions with applications, ICML 2017. [2] E. R. Elenberg, R. Khanna, A. G. Dimakis, and S. Negahban. Restricted strong convexity implies weak submodularity, https://arxiv.org/abs/1612.00804

Reviewer 2



The paper analyzed the performance of the greedy algorithm and POSS algorithm under the noise. Moreover, the paper proposed PONSS, a modification of POSS, that has better performance guarantee than POSS. Theorem 5 (performance analysis of PONSS) seems very interesting. However, I did not understand the correctness. Thus I tentatively set my decision "wrong". Please clarify this point. I checked other proofs and did not find serious flaw. [Concern] First, I would like to clarify the problem setting. Can we obtain sufficiently accurate estimation of f(x) by evaluating F(x) many times? If NO, the proof of the correctness of Algorithm 3 (PONSS) will be incorrect because the events "F(\hat x) \le F(x)" are dependent, so "Pr(\hat x \in argmin F) \le (1/2)^B" does not hold. --- [After rebuttal] I believe the paper is wrong. The most serious point is line 210: "We know from Eq. (5) that Pr(F(\hat x) \le F(x)) \le 0.5." To derive Eq.(5), we must ensure that the comparisons are independent. However, if we have values of F(x), this is wrong. For example, if F(x) = f(x) + (0 with prob 1/2, M with prob 1/2) for large M, we have "Pr(F(\hat x) \le F(x)) \le 0.5" but Eq.(5) is not hold. I asked this point in the review, and the authors' answered that we can evaluate the function multiple times. However, this causes another serious issue: "argmin" in line 9 of PONSS algorithm does not make sense, because the comparisons are inconsistent. Moreover, if this issue is solved, the comparison with other algorithms does not make sense because the only PONSS follows the different computational model.